# Separate the Wheat from the Chaff: Winnowing Down Divergent Views in Retrieval Augmented Generation

## Abstract

Retrieval-augmented generation (RAG) enhances large language models (LLMs) by integrating external knowledge sources to address their limitations in accessing up-to-date or specialized information. A natural strategy to increase the likelihood of retrieving relevant information is to expand the number of retrieved documents. However, involving more documents could introduce significant noise, as many documents may be irrelevant or misleading, thereby reducing the overall accuracy of the generated responses. To overcome the challenge associated with handling a larger number of documents, we propose WinnowRAG, a novel RAG framework designed to systematically filter out noisy documents while preserving valuable content – a process we refer to as **winnowing**. WinnowRAG operates in two stages: In Stage I, we perform **query-aware clustering** to group similar documents and form distinct topic clusters. Each cluster is assigned to an LLM agent for generating a unique answer. In Stage II, we perform winnowing, wherein a critic LLM evaluates the outputs of multiple agents and iteratively separates useful documents from noisy ones. To retain useful documents when discarding agents, we propose two **strategic merging** techniques to ensure that only relevant knowledge is used for generating the final response. Crucially, WinnowRAG is model-agnostic and does not require any model fine-tuning, making it easily adaptable to various tasks. Extensive experiments on various realistic datasets demonstrate the effectiveness of WinnowRAG over state-of-the-art baselines.

## 1 Introduction

Large language models (LLMs) have achieved significant success in various tasks such as text generation and question answering (Brown et al., 2020; Team et al., 2023; Dubey et al., 2024). While LLMs can store vast amounts of knowledge within their parameters, they exhibit weakness in specific knowledge-extensive tasks (Yoran et al., 2024). For example, when the input queries demand up-to-date information or out-of-domain knowledge, which is not present in the pre-training corpus (Shuster et al., 2021), LLMs would struggle to provide accurate answers (Zhang et al., 2023).

To overcome limitations in handling knowledge-intensive tasks, retrieval-augmented generation (RAG) has been proposed to improve LLMs by integrating external knowledge sources (Asai et al., 2023b; Zhao et al., 2024). Specifically, RAG retrieves relevant documents from external sources and incorporates them into the LLM's input, in order to help LLMs generate accurate responses in knowledge-intensive tasks (Yu et al., 2023). Consequently, RAG could benefit from the vast and consistently updated knowledge base to provide factual and timely knowledge. RAG frameworks typically retrieve multiple documents to ensure the inclusion of relevant information (Petroni et al., 2021). However, this approach can also introduce irrelevant or incorrect documents, which may hinder the LLM's ability to extract accurate information (Jiang et al., 2023; Jin et al., 2024).

In practice, retrieving more documents does not necessarily improve the RAG performance. As shown in Fig. 1, increasing the number of retrieved documents raises the probability that the correct information is included – enhancing the recall rate. However, beyond a certain threshold, adding more documents introduces significant noise, which can negatively impact the accuracy of the final answer. This presents the challenge in handling large sets of documents: while involving more docu-

ments may have a theoretically higher upper bound of accuracy, it simultaneously introduces greater challenges in processing them effectively. This trade-off explains why most existing approaches limit the number of retrieved documents to fewer than 20 (Wei et al., 2024; Wang et al., 2024c).

In this work, we propose to leverage large sets of retrieved documents by strategically filtering out noisy ones while retaining those that are useful, a process we refer to as **winnowing**. ❶ To handle a large number of documents, we first introduce **query-aware clustering**, which groups documents based on similar perspectives or information related to the query. This allows us to identify a range of topics within the retrieved documents, enabling filtering at the topic level rather than processing each document individually. This design significantly improves efficiency. Moreover, each cluster is assigned an LLM agent to provide a cluster-specific answer. ❷ To avoid discarding useful information, we propose a strategic, **merging-based winnowing** approach that filters out noisy documents while selectively retaining relevant ones. In particular, only a subset of documents from

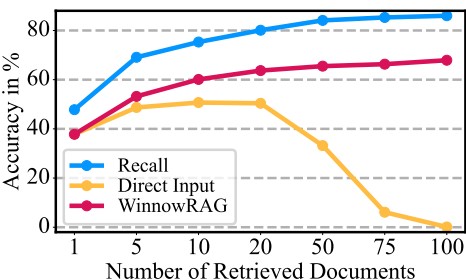

Figure 1: The accuracy results of the recall (i.e., upper bound), direct input, and WinnowRAG on the NaturalQ (Kwiatkowski et al., 2019) dataset with different numbers of retrieved documents.

each cluster is discarded, allowing us to refine the information extracted from a large document set. Throughout the winnowing process, we employ a critic LLM to evaluate the noisiness of answers generated from document clusters and guide the filtering process. Additionally, WinnowRAG requires no task-specific supervision, relying solely on a multi-agent framework with pretrained LLMs. Without any additional tuning, WinnowRAG can be easily adapted to a wide range of tasks. Our contributions are summarized as follows:

- **Framework:** We introduce WinnowRAG, a novel retrieval-augmented generation framework that clusters documents by topic and progressively filters out irrelevant or noisy documents using LLM agents. This structured filtering enhances the quality of the retrieved information.

- **Innovation and Adaptability:** WinnowRAG leverages the increased number of retrieved documents while minimizing the influence of irrelevant or incorrect content through its filtering (i.e., winnowing) mechanism. Notably, it operates without task-specific supervision, utilizing a multi-agent approach with pretrained LLMs. This eliminates the need for fine-tuning, making it versatile and easily applicable to a wide range of tasks.

- **Experiments and Results:** Through extensive experiments, we show that WinnowRAG consistently outperforms existing retrieval-augmented generation methods on several knowledge-intensive tasks. These results highlight its effectiveness in managing noisy data and boosting the performance of retrieval-augmented generation.

## 2 RELATED WORK

**Retrieval Augmented Generation.** Large language models (LLMs) struggle with domain-specific or knowledge-intensive tasks (Kandpal et al., 2023), often producing "hallucinations" (Zhang et al., 2023) when dealing with queries outside their training data or requiring up-to-date information. Retrieval-Augmented Generation (RAG) addresses this by retrieving relevant documents from external knowledge bases, reducing the risk of generating incorrect content (Lewis et al., 2020; Izacard & Grave, 2020; Asai et al., 2023a; Borgeaud et al., 2022; Guu et al., 2020; Gao et al., 2023). Recent works have primarily focused on enhancing precision and recall while minimizing irrelevant or toxic outputs that compromise the quality and reliability of responses (Shi et al., 2024; Ma et al., 2023; Jiang et al., 2023; Baek et al., 2023; Xu et al., 2023; Shi et al., 2024; Wang et al., 2024b; Luo et al., 2023). Among them, Self-Reflective RAG (Asai et al., 2023b) fine-tunes a general-purpose LLM to generate specific tags for self-reflection. Speculative RAG (Wang et al., 2024c) adopts instruction-tuned LLMs as drafters to offer diverse perspectives while reducing input token counts per draft. Moreover, InstructRAG (Wei et al., 2024) applies self-synthesized rationales as supervised fine-tuning data to train the model. However, these approaches require prior task-specific knowledge and additional instruction-tuning of LLMs, which is resource-intensive and limits their adaptability

across different domains. In contrast, we harness the potential of LLMs by assigning documents to agents and filtering out irrelevant content within a multi-agent winnowing framework. Our proposed method, WinnowRAG, is highly adaptable across domains without requiring task-specific signals or additional fine-tuning.

**LLMs as Critics.** Similar to humans, LLMs exhibit the ability to provide natural language feedback or critique, either based on their own internal knowledge (Wang et al., 2023; Zheng et al., 2024) or by utilizing external tools (Gao et al., 2022; Gou et al., 2023). Previous research has primarily focused on using such critiques to refine and improve the model's initial outputs on its own (Madaan et al., 2024; Shinn et al., 2024), or in multi-agent frameworks through discussion (Lu et al., 2024; Wang et al., 2024a; Chen et al., 2023) and debate (Du et al., 2023; Michael et al., 2023; Xiong et al., 2023; Khan et al., 2024; Subramaniam et al., 2024). To the best of our knowledge, RA-ISF (Liu et al., 2024) has the most similar framework design to ours in the field of RAG by utilizing self-feedback to iteratively filter out irrelevant retrieved documents. However, while RA-ISF focuses on denoising through query decomposition, our method directly filters the initial documents using a multi-agent framework. In our approach, LLM agents are assigned different groups of documents to form various perspectives. During inference time, a critic LLM progressively identifies agents with irrelevant or harmful content, enabling explicit denoising of the retrieved information with natural language feedback and reducing the risk of generating incorrect or misleading outputs.

## 3 METHODOLOGY

In this section, we first formulate the problem setting in Section 3.1 before introducing the proposed framework, WinnowRAG, which effectively filters irrelevant documents without relying on task-specific knowledge. As illustrated in Figure 2, WinnowRAG operates through two stages: query-aware clustering (Stage I) and multi-agent winnowing (Stage II). In Stage I (§ 3.2), the retrieved external documents are clustered into groups based on their perspectives relevant to the query, with each group assigned to an LLM agent. In Stage II (§ 3.3), agents with similar perspectives are merged to form super-agents, consolidating their respective documents. These super-agents then participate in a multi-round reflection process, called winnowing, where a critic LLM provides feedback to refine the results while filtering out irrelevant information. During each round, the critic LLM evaluates the agents' responses. Agents that are producing misleading outputs, from the critic LLM's perspective, will be merged with the remaining agents. A key challenge in both merging processes is to balance the inclusion of relevant documents while eliminating noise. To address this, we leverage the embedding space and design two merging methods, as detailed in Section 3.4.

### 3.1 PROBLEM FORMULATION

We follow the standard RAG setting (Wei et al., 2024; Asai et al., 2023b), where each task $\mathcal{T}$ consists of a triple $(\mathcal{Q}, \mathcal{A}, \mathcal{D})$. Given a question-answer pair $(q, a) \in (\mathcal{Q}, \mathcal{A})$, a retriever $\mathcal{R}$ retrieves supporting documents $\mathcal{D}_{\mathcal{R}} \subseteq \mathcal{D}$ from the external knowledge base $\mathcal{D}$. We aim to filter out noisy documents in $\mathcal{D}_{\mathcal{R}}$ such that the LLM can better generate the response $a'$ containing the correct answer based on the retrieved external knowledge, i.e., to maximize $\mathbb{E}_{(q,a)}\mathcal{M}(a, a')$, where $\mathcal{M}$ represents a specific evaluation metric, e.g., *accuracy*.

### 3.2 STAGE I: QUERY-AWARE CLUSTERING

In this section, we provide a detailed explanation of the query-aware document clustering process. The key motivation is that the external documents often contain diverse and noisy content (Wang et al., 2024c). By clustering the documents based on their relevance to the query, each group will have a more consistent perspective regarding the query. This enables each LLM agent to provide relatively consistent answers when using a specific group of documents as input. Specifically, we first cluster the retrieved documents into groups using query-aware embeddings and the $K$-Means clustering algorithm (Anderberg, 2014).

To ensure documents with similar perspectives on a query are grouped together, given a query $q$ and a set of retrieved documents $\mathcal{D}_{\mathcal{R}} = \{d_1, d_2, \ldots, d_N\}$ from the external database, we encode each document alongside the query using a structured prompt. This representation is then processed using the $K$-Means algorithm to group documents with related viewpoints. The clustering is performed

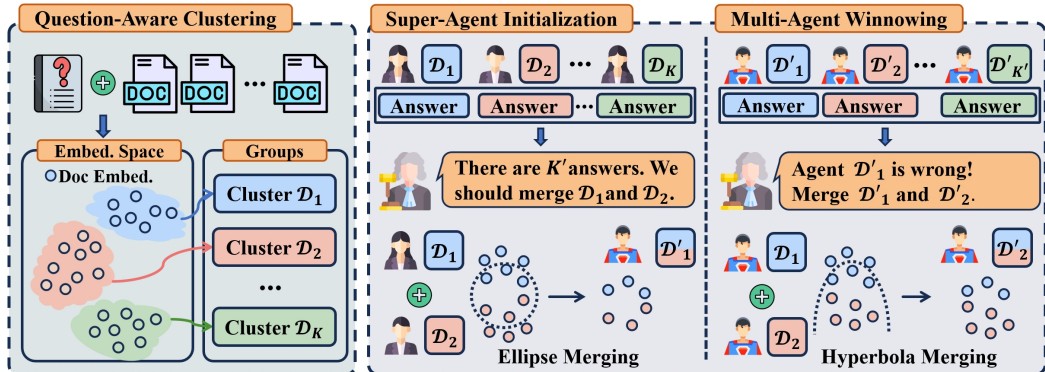

Figure 2: The overall process of our WinnowRAG framework. We first perform query-aware clustering to group documents with similar semantic meanings with respect to the query. In Stage II, we first perform agent initialization to form multiple super-agents that will be used in the following winnowing steps. During multi-agent winnowing, we gradually discard agents with incorrect answers, guided by the critic LLM, while retaining useful documents.

as follows:

$$\text{emb}(d_i) = f(\texttt{Prompt}(q \oplus d_i)), \quad i = 1, 2, \ldots, N.$$
$$\{\mathcal{D}_1, \mathcal{D}_2, \ldots, \mathcal{D}_K\} = \texttt{K-Means}(\text{emb}(d_1), \text{emb}(d_2), \ldots, \text{emb}(d_N)). \tag{1}$$

Here $f(\cdot)$ represents the text embedding model (e.g., Sentence-BERT (Reimers, 2019)); $\text{emb}(d_i)$ is the query-aware embedding for the document $d_i$; $\mathcal{D}_j$ is a cluster of documents with similar contents; $K$ is a hyper-parameter that controls the number of clusters. We then assign each document group $\mathcal{D}_j$ to an LLM agent $A_j \in \{A_1, A_2, \ldots, A_K\}$, which is a general pretrained LLM. At this stage, we typically use a relatively large value of $K$ (e.g., $K = 10$) to ensure that different clusters contain divergent views. Agents assigned to a noisy cluster will produce responses that deviate from the correct answer, making it easier to identify and eliminate them in the subsequent winnowing stage.

### 3.3 STAGE II: MULTI-AGENT WINNOWING

▷ **Super-Agent Initialization.** To remove redundant agents and reduce further winnowing rounds for efficiency, we first query the agents from Stage I to provide answers to the query based on their assigned documents (prompt provided in Appendix B.1). Next, we introduce a critic LLM, which is a pretrained language model, to summarize the distinct responses from them *without* making judgments (prompt provided in Appendix B.3). We then merge any pair of agents with similar answers into a super-agent. When merging, our goal is for the super-agent to retain documents that adequately represent the perspectives of both original agents. To achieve this, we operate in the embedding space and propose the **Ellipse Merging** strategy. Intuitively, when two agents arrive at similar conclusions, their document embeddings should be closer. We define an ellipse in the embedding space, with its foci close to the centroids of the two agents' document embeddings, and select the documents within the ellipse as documents for the super-agent. In Section 3.4, we introduce the ellipse merging process in detail.

▷ **Multi-Agent Winnowing.** After the super-agent initialization process, we have $K'$ super-agents $\mathcal{A}' = \{A'_1, A'_2, \ldots, A'_{K'}\}$, where $K'$ is the number of distinct responses determined by the critic LLM and $K' \leq K$. Each super-agent $A'_j$ now has a different perspective from others to the query. We then propose the multi-agent winnowing stage to harness the critic LLM's ability to identify potential errors in the super-agents' outputs, thereby producing more consistent and precise answers.

In multi-agent winnowing, we perform maximally $M$ rounds of winnowing. During each round of winnowing, the super-agents act in parallel, each presenting an argument based on the critic LLM's feedback (from the previous round) and its current documents. To provide enough supportive information to the critic LLM to make decisions, each argument includes three components: (a) *evidence*, extracted from the documents of that agent, (b) *rationale*, explaining how the evidence supports the

conclusion, and (c) the *final answer*. The detailed prompt is provided in Appendix B.2. The critic LLM oversees and manages the entire winnowing process by taking one of the following actions: (a) concluding the winnowing and obtaining the final answer $a'$, or (b) continuing the winnowing by identifying incorrect super-agents, denoted as $\mathcal{A}'_{inc}$. If the winnowing process concludes, the critic LLM will output the final answer $a'$. If the critic LLM decides to continue, each super-agent $A'_j$ in $\mathcal{A}'_{inc}$ is merged with the closest remaining agent $A'_i$, i.e.,

$$A'_i = \underset{A'_k \in \mathcal{A}' \setminus \mathcal{A}'_{inc}}{\arg\min} |\mu'_i - \mu'_k|, \tag{2}$$

where $\mu'_k$ is the centroid of the super-agent $A'_k$'s document embeddings.

When merging the incorrect super-agent $A'_j$ with a remaining agent $A'_i$, our goal is to retain helpful documents from $A'_j$'s documents while preventing noisy ones from being assigned to $A'_i$ for the next round of winnowing. To achieve this, we propose the **Hyperbola Merging** strategy. Specifically, we define a hyperbola in the embedding space, using the foci close to the centroids of the two super-agents' document embeddings, $\mu'_i$ and $\mu'_j$. Document embeddings that fall on the opposite side of the hyperbola relative to $\mu'_j$ will have a smaller distance to $\mu'_i$ by a fixed threshold. Assigning these documents to $A'_i$ for the next round of winnowing ensures a more specialized and complementary merging process while explicitly filtering out noisy documents. We describe this hyperbola merging process in detail in Section 3.4.

After each round, the rationales provided by the critic LLM will be handed over to each remaining super-agent. The detailed prompt is provided in Appendix B.4. Notably, this enables the super-agents to incorporate feedback from the previous round and generate improved responses in the subsequent round.

### 3.4 MERGING STRATEGIES

Stage II involves two types of agent merging processes. During the initialization of super-agents, we focus on merging agents with similar views, while in the winnowing process, incorrect super-agents are merged into the remaining ones. Both processes require balancing the inclusion of relevant documents with the elimination of noise. To address this challenge, we propose two merging strategies in the embedding space.

▷ **Ellipse Merging.** This strategy is used to merge agents with similar answers in the super-agent initialization step. We denote the $K$ agents as $\{A_1, A_2, \ldots, A_K\}$, and their corresponding documents as $\{\mathcal{D}_1, \mathcal{D}_2, \ldots, \mathcal{D}_K\}$.

Suppose that the answer of agent $A_i$ is sufficiently similar to that of $A_j$, decided by the critic LLM. We aim to merge these two agents by merging the documents of these two agents, i.e., $\mathcal{D}_i$ and $\mathcal{D}_j$. Intuitively, since these two agents bear similar answers, their documents should also bear similar meanings. Therefore, to retain the documents that are mostly helpful, we propose to select documents that are close to both clusters. As such, we define the set of merged documents, $\mathcal{D}_{i,j}$, based on their distances to the centroids of cluster $\mathcal{D}_i$ and $\mathcal{D}_j$ as follows:

$$\mathcal{D}_{i,j} = \{x \mid d_{A_i}(x) + d_{A_j}(x) \le T_{ij}, x \in \mathcal{D}_i \cup \mathcal{D}_j\},$$

$$\text{where } T_{ij} = \frac{1}{|\mathcal{D}_i| + |\mathcal{D}_j|} \sum_{x \in \mathcal{D}_i \cup \mathcal{D}_j} \left( d_{A_i}(x) + d_{A_j}(x) \right), \text{ and } d_{A_i}(x) = ||\text{emb}(x) - \mu_i||_2. \tag{3}$$

Here $\mu_i$ is the centroid of the $i$-th cluster, i.e., $\mu_i = \frac{1}{|\mathcal{D}_i|} \sum_{x \in \mathcal{D}_i} \text{emb}(x)$. In the above equation, we set a threshold $T_{ij}$, such that the documents with a summed distance to centroids $\mu_i$ and $\mu_j$ less than $T_{ij}$ are included in the merged set. As a result, the documents that are included in this defined ellipse will be kept during merging. To determine the value of the threshold $T_{ij}$, we resort to selecting the summed distance to both centroids, averaged across documents in the two clusters. This describes the average summed distance of any document to both centroids. Thus, documents with a summed distance less than $T_{ij}$ are more likely to be close to both clusters.

▷ **Hyperbola Merging.** At the end of each winnowing round, we aim to merge the documents of two agents, one of which is considered incorrect by the critic LLM. Rather than selecting documents

Table 1: Dataset statistics and the corresponding retrieval models.

| Dataset | Train | Test | Retriever | Recall@5 | Recall@20 |
|---|---|---|---|---|---|
| Natural Questions | 79,168 | 3,610 | DPR | 68.8 | 80.1 |
| TriviaQA | 78,785 | 11,313 | Contriever | 73.5 | 82.7 |
| PopQA | 12,868 | 1,399 | Contriever | 68.7 | 78.2 |
| ASQA | 4,353 | 948 | GTR | 82.2 | 87.5 |
| 2WikiMQA | 167,454 | 12,576 | BM25 | 33.2 | 62.3 |

that are close to both clusters, as in Ellipse Merging, we now select documents that are close to the potentially correct agent while sufficiently far from the other. This strategy helps in identifying documents that are more likely to be helpful but clustered into the incorrect agent.

Suppose that super-agent $A_i$ is considered potentially correct, and another super-agent $A_j$ is considered incorrect. We aim to merge their documents in a way that emphasizes documents that are close to $A_i$ but distant from $A_j$. Nevertheless, even though $A_j$ provides a wrong answer, the documents in $\mathcal{D}_j$ may still be useful for reasoning of subsequent steps. Therefore, we aim to keep most documents of agent $A_i$ while only keeping the documents of $A_j$ that are close to $A_i$. Therefore, we propose the merging conditions as follows:

$$\begin{cases} d_{A_i}(x) < T_i, \\ d_{A_j}(x) > T_j, \end{cases} \tag{4}$$

where $d_{A_i}(x) = ||\text{emb}(x) - \mu_i||_2$ and $d_{A_j}(x) = ||\text{emb}(x) - \mu_j||_2$ represent the distances of a document $x$ to the centroids of the clusters associated with agents $A_i$ and $A_j$, respectively. The value $T_i$ is selected as a threshold below which documents are considered close to the centroid of agent $A_i$, while $T_j$ is the threshold above which documents are considered distant from agent $A_j$. Combining the merging conditions, the set of merged documents, $\mathcal{D}_{i,j}$, is achieved as follows:

$$\mathcal{D}_{i,j} = \{x \mid d_{A_j}(x) - d_{A_i}(x) > T_j - T_i, \ x \in \mathcal{D}_i \cup \mathcal{D}_j\},$$
$$T_i = \frac{1}{|\mathcal{D}_i| + |\mathcal{D}_j|} \sum_{x \in \mathcal{D}_i \cup \mathcal{D}_j} d_{A_i}(x), \quad T_i = \frac{1}{|\mathcal{D}_i| + |\mathcal{D}_j|} \sum_{x \in \mathcal{D}_i \cup \mathcal{D}_j} d_{A_j}(x). \tag{5}$$

Therefore, remained documents are included in a hyperbola defined by the above equation. This merging strategy helps in identifying and merging documents that are primarily relevant to agent $A_i$ but distant from agent $A_j$, allowing for a focused merging of contrasting perspectives (of $A_i$ and $A_j$). By applying this hyperbola-based merging criterion, we highlight documents that contribute to divergent views, ensuring a more specialized and complementary merging process.

## 4 EXPERIMENTS

### 4.1 DATASETS

In our experiments, we utilize public RAG benchmarks: NaturalQ (Kwiatkowski et al., 2019), TriviaQA (Joshi et al., 2017), PopQA (Mallen et al., 2023), ASQA (Stelmakh et al., 2022), and 2WikiMQA (Ho et al., 2020). Detailed statistics for the datasets are provided in Table 1. We utilize the Wikipedia corpus as the retrieval source and evaluate our approach using both sparse and dense pre-trained retrievers, such as BM25 (Robertson & Walker, 1994), DPR (Karpukhin et al., 2020), GTR (Ni et al., 2022), and Contriever (Izacard et al., 2021). Retrieval performance is assessed by Recall@5 and Recall@20, which checks if the top 5 or 20 retrieved documents include the correct answer. In line with established evaluation protocols (Asai et al., 2023b; Wei et al., 2024), we use Exact Match (EM) for ASQA (Stelmakh et al., 2022). For the other datasets, we consider accuracy, which measures whether the generated model outputs include the correct ground-truth answers (Mallen et al., 2023; Schick et al., 2024).

Table 2: The overall results of our framework and baselines on five downstream tasks with and without fine-tuning the LM. The best performance is shown in **bold**. "–" denotes that the results are not reported in the original work or are not applicable. We report the *accuracy* for datasets NQ, TriviaQA, PopAQ, and 2WikiMQA, and report the *exact match* for dataset ASQA. "8B", and "70B" represent Llama-3-8B-Instruct, and Llama-3-70B-Instruct, respectively.

| Dataset | PopQA | | TriviaQA | | NQ | | 2WikiMQA | | ASQA | |
|---|---|---|---|---|---|---|---|---|---|---|
| Llama w/o Fine-tune | 8B | 70B | 8B | 70B | 8B | 70B | 8B | 70B | 8B | 70B |
| **Zero-shot Prompting** | 22.8 | 28.9 | 69.4 | 80.6 | 46.6 | 57.9 | 45.6 | 57.5 | 30.6 | 39.1 |
| **In-Context RALM** | 62.3 | 63.8 | 71.4 | 76.3 | 56.8 | 60.2 | 43.4 | 51.2 | 40.0 | 43.1 |
| **ICL** | 63.1 | 63.9 | 74.2 | 79.1 | 60.1 | 62.9 | 45.3 | 53.9 | 42.6 | 45.4 |
| **InstructRAG-ICL** | 64.2 | 65.5 | 76.8 | 81.2 | 62.1 | 66.5 | 50.4 | 57.3 | 44.7 | 47.8 |
| **WinnowRAG** | **68.1** | **68.8** | **79.3** | **81.6** | **66.8** | **68.3** | **56.3** | **58.4** | **47.9** | **48.5** |
| Llama w/ Fine-tune | 8B | 70B | 8B | 70B | 8B | 70B | 8B | 70B | 8B | 70B |
| **SFT** | 61.0 | – | 73.9 | – | 56.6 | – | 56.1 | – | 43.8 | – |
| **Self-RAG** | 55.8 | – | 71.4 | – | 42.8 | – | 32.9 | – | 36.9 | – |
| **RetRobust** | 56.5 | – | 71.5 | – | 54.2 | – | 54.7 | – | 40.5 | – |
| **InstructRAG-FT** | 66.2 | – | 78.5 | – | 65.7 | – | 57.2 | – | 47.6 | – |

## 4.2 BASELINES

In this subsection, we introduce the baseline used in our experiments for comparison. Specifically, we evaluate our approach against a variety of RAG baselines, considering settings with and without training. For baselines with training, we consider ❶ Supervised Fine-tuning (SFT), which optimizes the likelihood of generating the correct answer; ❷ RetRobust (Yoran et al., 2024), which fine-tunes the model by incorporating both relevant and irrelevant contexts to improve robustness; ❸ Self-RAG (Asai et al., 2023b), which adjusts retrieval using special reflection tokens; and ❹ InstructRAG (Wei et al., 2024), which instructs the LM to provide rationales used for fine-tuning. Notably, for RetRobust and Self-RAG, we adopt their results with Llama-3-Instruct-8B as the backbone model, as reported in InstructRAG, instead of using Llama-2 in the original papers. For baselines without training, we consider ❶ In-context Retrieval-Augmented Language Modeling (RALM) (Ram et al., 2023), a prompting technique that enhances the non-retrieval baseline by providing the model with relevant documents; and ❷ In-context Learning (ICL), which uses ground-truth question-answer pairs from the training set as demonstrations, and ❸ Zero-shot Prompting, which directly queries LLMs for the answer.

## 4.3 RETRIEVAL SETUP

Following Self-RAG (Asai et al., 2023b) and InstructRAG (Wei et al., 2024), we perform retrieval from documents in the Wikipedia dump in DPR (Karpukhin et al., 2020) for all datasets. Moreover, each document is a separate text extracted from Wikipedia articles, containing up to 100 words. Regarding the specific retrievers, we employ Contriever-MS MARCO for PopQA and TriviaQA and DPR for Natural Questions. For datasets ASQA and 2WikiMultiHopQA, we use GTR and BM25, respectively. By default, we retrieve the top 50 documents for all tasks. For the dense retrievers, we utilize their official weights. For the sparse retriever BM25, we implement it using Pyserini (Lin et al., 2021).

## 4.4 INFERENCE DETAILS

Our experiments are all conducted on four Nvidia A100 GPUs, each with 80GB of memory. To facilitate the inference process, we utilize the vLLM pacakge (Kwon et al., 2023). Greedy seconding is applied for inference. We set $K = 10$ for our framework and the maximum token length for all models as 4096. For the critic LLM, we use the same model as the agents. For ICL and InstructRAG-ICL, we follow InstructRAG and set the number of demonstrations as 2. Our code is provided at https://anonymous.4open.science/r/WinnowRAG-09B2/README.md.

## 4.5 COMPARATIVE RESULTS

In this subsection, we study the comparative results of our framework and other state-of-the-art RAG methods with and without training (or fine-tuning). Particularly, we present the results for RAG baselines without training using Llama-3-Instruct-8B and Llama-3-Instruct-70B. For RAG methods with training, we consider Llama-3-Instruct-8B, as the fine-tuning results on Llama-3-Instruct-70B are difficult to obtain and not reported in existing works. The results are presented in Table 2. Comparing the results of RAG baselines without training, we can observe that ❶ **Parameter Size Matters.** All methods present better results with a larger model parameter size, which increases from 8B to 70B. This demonstrates that when not fine-tuned, a larger LM could potentially provide better reasoning capability to utilize the retrieved documents for answering. Notably, WinnowRAG achieves superior performance even with the smaller model Llama-3-Instruct-8B. This indicates that WinnowRAG does not require powerful LLMs to function, thereby leading to better practicability. ❷ **Retrieval Helps.** In-context RALM, ICL, and InstructRAG-ICL generally outperform the zero-shot prompting method, which does not involve any retrieval. This indicates that for such open-domain question-answering tasks, the involvement of retrieved documents is crucial. ❸ **Outstanding Performance.** Our framework consistently outperforms all other training-free baselines across various datasets. Particularly, WinnowRAG is particularly superior on datasets PopQA and NQ with lower Recall@20 in comparison to TriviaQA and ASQA. This demonstrates WinnowRAG's ability to effectively filter and refine retrieved documents, even in scenarios where the correct information may be distributed across multiple noisy sources. ❹ **Model-Agnostic Capabilities.** One of the key insights from these experiments is the model-agnostic nature of WinnowRAG. Despite the use of smaller models like Llama-3-8B-Instruct, our framework demonstrates the ability to achieve better performance compared to larger fine-tuned models on four datasets. This adaptability makes WinnowRAG, a training-free framework, highly practical for deployment in scenarios where computational resources are limited, or where large-scale fine-tuning is not feasible. The fact that WinnowRAG achieves superior results without requiring task-specific training further underscores its flexibility and broad applicability.

## 4.6 ABLATION STUDY

In this subsection, we conduct experiments while removing specific modules of our framework to separately study their effects on the performance. Particularly, we consider the following variants of our frameworks: ❶ We remove the query-aware clustering during Stage I and replace it with random splitting. We refer to this variant as WinnowRAG\Q. ❷ We remove the strategic merging techniques during Stage II. In this variant, when merging agents with the same answers, we randomly keep half of the documents of both agents and combine them into one agent. When merging agents with different answers, we directly discard all documents of the agent with a wrong answer.

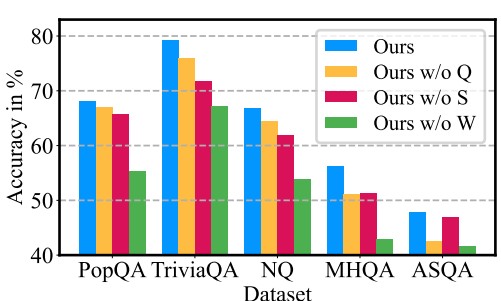

Figure 3: The ablation study results of WinnowRAG on five datasets.

We refer to this variant as WinnowRAG\S. ❸ We remove the entire multi-agent winnowing module, i.e., Stage II, and directly select one answer from responses of all clusters using the critic LLM. We refer to this variant as WinnowRAG\W. From the results presented in Fig. 3, we can obtain the following observations: ❶ **WinnowRAG\Q results in a moderate drop in performance.** This can be attributed to the loss of grouping based on document content, which undermines the framework's ability to effectively cluster related information. Random splitting leads to a less coherent selection of documents, increasing the noise in agent responses and reducing the critic's ability to accurately assess the outcome of each cluster. ❷ **WinnowRAG\S shows that the strategic merging techniques are critical**, particularly in datasets with a high recall rate like NQ and TriviaQA. Without merging strategies, the framework struggles to retain useful documents. Randomly discarding documents or entirely removing those from agents introduces more noise and leads to suboptimal performance, as relevant information may be inadvertently lost. ❸ **WinnowRAG\W, results in the largest performance drop.** This

suggests that the multi-agent winnowing process plays a fundamental role in our framework. The absence of iterative winnowing leads to a lack of thorough evaluation of the agents' responses, and the critic LLM alone is insufficient to make optimal selections from a large set of noisy or conflicting responses. This variant highlights how crucial multi-agent winnowing is in ensuring that only the most relevant and accurate documents contribute to the final answer.

### 4.7 PARAMETER SENSITIVITY

In this subsection, we explore the sensitivity of our proposed framework WinnowRAG to several key parameters. These experiments aim to understand the impact on the final model performance by varying ❶ the rounds of winnowing, ❷ the number of retrieved documents, and ❸ the number of query-aware clusters. We choose to adjust these parameters can as they can affect both the quality and efficiency of the retrieval-augmented generation process.

**Rounds of Winnowing.** An essential aspect of our framework is the number of winnowing rounds used in the multi-agent winnowing process. During each round, super-agents engage in a structured discussion, iteratively refining their responses and converging towards the most accurate answer, with noisy or incorrect agents gradually being filtered out. To understand the sensitivity of performance to the number of winnowing rounds, we conduct experiments where the winnowing process was termi-

Table 3: Performance of WinnowRAG with different rounds of winnowing.

| Dataset | PopQA | TriviaQA | NQ | MHQA | ASQA |
|---------|-------|----------|------|------|------|
| $M = 1$ | 62.5 | 74.2 | 60.3 | 50.1 | 43.2 |
| $M = 2$ | 65.7 | 78.9 | 63.4 | 53.2 | 44.9 |
| $M = 3$ | 68.1 | 79.3 | 66.8 | 56.3 | 47.9 |
| $M = 4$ | 69.2 | 79.5 | 67.4 | 57.0 | 47.7 |
| $M = 5$ | 68.5 | 79.4 | 67.2 | 56.8 | 46.8 |

nated at different rounds, observing the effects on the final output. From the results presented in Table 3, we can observe several trends: ❶ **Early stopping yields suboptimal results.** Terminating the winnowing process after just 1 or 2 rounds leads to suboptimal answers. This is because the early rounds of winnowing often do not provide sufficient time for the agents to fully resolve conflicts or eliminate noisy contributions. In these early rounds, agents may still involve irrelevant documents, which hinders the ability of the critic LLM to derive a well-informed final answer. ❷ **More rounds may not always help.** While additional rounds of winnowing help improve the accuracy by progressively refining the answers, our results show that after a certain threshold, further iterations lead to decreasing performance. Beyond this point, the performance slightly degrades. This decline can be attributed to the unnecessary complexity introduced by excessively extending the winnowing process. As the winnowing continues, the growing complexity can make it more difficult for the critic LLM to track critical information. Misinterpretations or misunderstandings may occur, leading to degraded decision-making or incorrect conclusions. ❸ **Optimal numbers of rounds may differ.** The results suggest that there is an optimal number of winnowing rounds where the balance between refinement and complexity is achieved. In this case, the framework has effectively filtered out noisy agents and converged on the most relevant information without incurring the risks of filtering out useful documents. Notably, determining this optimal number is task-dependent. For example, the performance on dataset TriviaQA stabilizes earlier, due to its simplicity, while other datasets generally require more rounds.

**Number of Retrieved Documents.** The number of documents retrieved for each query is critical, as more documents can provide additional relevant information but may also introduce noise. In Fig. 4, we present the results by varying the number of retrieved documents. We observe that: ❶ Retrieving fewer documents (e.g., 10 or fewer) may result in the model missing important information, as the necessary knowledge for answering the question may not be sufficiently covered. This could lead to a lower accuracy due to insufficient evidence available to the agents. ❷ Increasing the number of retrieved documents can improve the quality of the answer by providing a richer

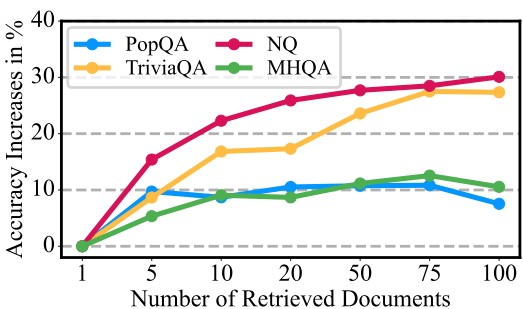

Figure 4: The accuracy improvement (over using one retrieved document) results of WinnowRAG with different numbers of retrieved documents.

knowledge source and increasing the chances of capturing relevant information. However, retrieving too many documents could overwhelm the system with irrelevant information, introducing more noise and potentially harming the performance. Nevertheless, our framework exhibits further improved performance, demonstrating the robustness of our design against noise.

**Number of Query-Aware Clusters.** The number of query-aware clusters in Stage I, i.e., $K$, plays a significant role in the framework's ability to cover diverse perspectives or sets of information from the retrieved documents, as each agent could provide a potentially unique answer based on its assigned cluster of documents. Since the result of varying $K$ is tightly associated with the number of retrieved documents $N$, we hereby study the joint impact of both $K$ and $N$. Particularly, we conduct experiments by varying both of them on the dataset NatrualQ. It is noteworthy that $N \leq K$, otherwise the clustering becomes infeasible. From the results presented in Fig. 5. The key observations include:
❶ **Fewer clusters lead to poor performance.** When the number of clusters $(K)$ is too small, the framework's ability to cover diverse perspectives is significantly hindered. For example, the results with $K = 5$ are generally worse than

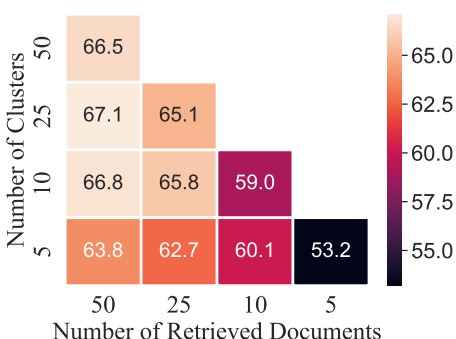

Figure 5: The results of WinnowRAG on dataset NaturalQ with varying numbers of query-aware clusters and retrieved documents.

the results with $k = 10$. Notably, with fewer clusters, each agent is forced to handle a broader range of documents, many of which may contain conflicting or irrelevant information. This reduces the precision of the generated answers, and thus the critic LLM struggles to resolve these conflicts, leading to suboptimal performance. This effect is particularly noticeable when the number of retrieved documents is large, as the few clusters cannot adequately filter and partition the information. ❷ **Too many clusters can also be detrimental.** Conversely, increasing the number of clusters beyond a certain point also results in performance degradation. For example, when the number of retrieved documents is 25 or 50, enlarging the number of clusters $K$ to 25 or 50 could impact the performance when compared to the results with $K = 10$. While more clusters allow agents to specialize in narrower sets of documents, excessive partitioning dilutes the amount of relevant information available to each agent, causing the loss of useful context. Additionally, when $K$ is high, the critic LLM must process a larger number of agents, adding unnecessary complexity to the winnowing process without corresponding gains in accuracy. ❸ **More retrieved documents require more clusters.** As the number of retrieved documents increases, the optimal number of clusters also needs to increase. For example, the best performance with $N = 25$ and $N = 50$ is achieved when $K = 10$ and $K = 25$, respectively. This is because when more documents are retrieved, they are likely to contain a wider range of information, both relevant and irrelevant. If the number of clusters remains small while the number of retrieved documents increases, the framework becomes overwhelmed by noise, reducing the accuracy of the final answers. Nevertheless, when the number of clusters $K$ is appropriately scaled with the number of retrieved documents, the agents can more effectively handle the information, leading to better overall performance.

## 5 CONCLUSION

In this work, we propose WinnowRAG, a novel training-free framework that effectively addresses the inherent challenges of utilizing a large number of retrieved documents in RAG systems. Specifically, with our designed stages of query-aware clustering and multi-agent winnowing, WinnowRAG manages to filter out noisy information in retrieved documents while retaining useful documents. As a result, WinnowRAG enhances the accuracy and relevance of generated responses without necessitating model-specific fine-tuning. The strong performance exhibited in experiments underscores its potential as a robust approach for integrating external knowledge into language models, providing insights for more reliable and contextual knowledge-intensive applications in various domains.

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

## A  IMPLEMENTAION DETAILS

In this section, we provide more details of our implementation. Specifically, we set $K$, the number of clusters as 10, and the number of retrieved documents $N$ as 50. Note that 50 is larger than the size of retrieved documents in most existing works, such as 5 and 10 in InstructRAG (Wei et al., 2024). We use vLLM (Kwon et al., 2023) to facilitate the inference of all models. We set the batch size as 200, using 4 A100 GPUs, each with 80GB of memory. By default, we set the maximum round of winnowing as 3, although the framework may terminate the winnowing process at earlier rounds. For all LLMs, we set the temperature as 0 to keep consistency across runs.

## B  PROMPT TEMPLATES

In this section, we provide the detailed prompts in our implementation.

### B.1  STAGE I AGENT RESPONSE GENERATION

---
**Stage I Agent Response Generation**

**Input:** You are given the following documents.
Document [1] (Title: · · · ): {contents}
· · ·
Based on the provided information, answer the following question: {question}. You are strictly prohibited from generating the answer based on your own knowledge.

Directly output your answer without any additional explanation.

**Output:** {answer}

---

### B.2  STAGE II SUPER-AGENT RESPONSE GENERATION

---
**Stage II Super-Agent Response Generation**

**Input:** You are given the following documents.
Document [1] (Title: · · · ): {contents}
· · ·
Based on the provided information, answer the following question: {question}. You are strictly prohibited from generating the answer based on your own knowledge.

Your response should consist of three components:
1. Extract a portion of the provided documents that directly supports your answer to the question. The extracted information should be concise and free from irrelevant details, serving as the evidence for your answer.
2. Explain how the evidence supports your final answer.
3. Present your final answer.

Format your response as follows:

Evidence: [YOUR EVIDENCE]

Explanation: [YOUR EXPLANATION]

Answer: [YOUR FINAL ANSWER]

**Output:** {response}

---

## B.3 STAGE I CRITIC LLM AGENT ANSWER SUMMARIZATION

---

### Stage I Critic LLM Agent Answer Summarization

**Input:** You are given the following answers from $\{K\}$ agents to the question: {question}.
Answer [1]: Answer: {answer}
...
Your task is to summarize the $\{K\}$ answers and remove duplicates.

Your response should consist of two components:
1. Deduplicate the provided answers. Exact matching is not required; answers are considered duplicates if they have the same semantic meaning. Output a list of unique answers.
2. Explicitly indicate which answers are duplicates, along with their corresponding indices.

Format your response as follows:

Unique answers: [LIST OF UNIQUE ANSWERS]

Duplicate answers: [LIST OF DUPLICATE ANSWERS]

**Output:** {response}

---

## B.4 STAGE II CRITIC LLM ANSWER JUDGEMENT

---

### Stage II Critic LLM Answer Judgement

**Input:** You are provided with the following responses from $\{K'\}$ agents to the question: {question}. Each response contains an answer, supporting evidence from the provided documents, and an explanation of how the answer was derived.
Response [1]: Answer: {answer}; Evidence: {evidence}; Explanation: {explanation}
...
Based on your knowledge and the provided information, you are tasked with the following:
1. Identify the misleading responses from the $\{K'\}$ that result in incorrect answers.
2. Determine whether a consistent answer can be derived from the remaining potentially correct responses.

Your response should consist of three components:
1. The list of responses with incorrect answers. Output a list of response IDs.
2. Provide an explanation for why these responses are considered incorrect, and why the remaining responses are considered correct.
3. Indicate yes or no, depending on whether a consistent answer can be derived from the remaining responses. If yes, also provide the consistent answer.

Format your response as follows:

Incorrect answers: [LIST OF INCORRECT RESPONSE IDS]

Explanation: [YOUR EXPLANATION]

Consistent answer: [YOUR ANSWER, IF APPLICABLE]

**Output:** {response}

---

