# OpenReview forum: "Separate the  Wheat from the Chaff: Winnowing Down Divergent Views in Retrieval Augmented Generation"
_ICLR.cc/2025/Conference — ICLR 2025 Conference Withdrawn Submission_

### Official Review · Reviewer_nTYQ · 2024-10-21

**Soundness:** 2
**Presentation:** 3
**Contribution:** 2
**Rating:** 5
**Confidence:** 5

**Summary:**

This paper proposes WinnowRAG, a RAG framework to systematically filter out noisy documents while preserving valuable content. WinnowRAG first performs query-aware clustering to group documents with similar semantic meanings with respect to the query. Then,
it performs agent initialization to form multiple super-agents that will be used in the following winnowing steps. During multi-agent winnowing, we gradually discard agents with incorrect answers, guided by the critic LLM, while retaining useful documents. Experiments are performed on QA tasks and show WinnowRAG performs better than some baselines.

**Strengths:**

1. Introduce WinnowRAG, a retrieval-augmented generation framework that clusters documents by topic and progressively filters out irrelevant or noisy documents using LLM agents.

2. The presentation is good and easy to follow.

3. The experimental results show WinnowRAG outperforms many RAG baselines that need training.

**Weaknesses:**

1. The technical contribution is limited. WinnowRAG is more biased towards the application of retrieval-augmented generation in engineering, but lacks sufficient insights for research. Methods in WinnowRAG, such as query-aware clustering, multi-agent winnowing are so common in this area. I cannot find inspiring enough ideas for the research of  RAG from this paper.

2. Multi-clusters and multiple LLM-based agents, computing overhead is an important concern for WinnowRAG.

3. Some baselines that use additional filters to evaluate the utility of retrieved texts should be considered, such as CRAG [1].

[1] Corrective Retrieval Augmented Generation

**Questions:**

1. Can you provide the comparison of computing overhead of WinnowRAG and other baselines?

2. Can you provide more details of WinnowRAG, such as what LLM-based agents are you use?

---

### Official Review · Reviewer_gZcJ · 2024-11-01

**Soundness:** 3
**Presentation:** 4
**Contribution:** 3
**Rating:** 5
**Confidence:** 5

**Summary:**

This paper proposes WinnowRAG, a multi-agent RAG framework designed to systematically filter out noisy documents while preserving valuable content among retrieved documents. The evaluation process is based on some famous RAG post-training frameworks and simple prompting/ICL. The experiment shows that this framework increases the QA accuracy of RAG systems.

a brief summary of WinnowRAG:
1. cluster the retrieved document and use each cluster to generate an answer.
2. merge the clusters based on answers.

**Strengths:**

1. The main research question of this paper is indeed important 'How to filter out the noise context'.
2. The proposed method used multi-agent and critical LLM to enhance the robustness of the proposed framework.

**Weaknesses:**

While the authors present an interesting approach, the evaluation would be strengthened by including additional baselines for comparison. Because the benchmarked frameworks are not filter-focused. For example, self-RAG is mainly for judging 'When to retrieve'. The main objective of this paper is more similar to the Reranker.

Thus, I recommend adding the following two baselines:

1. a comparison with using SOTA Reranker: the main function of reranker is further filtering based on retrieved documents, which is the same goal of this paper.
2. a comparison with a single agent to filter noise context: LLM-based Ranker

**Questions:**

The question is the same with the weakness section.

---

### Official Review · Reviewer_nEmb · 2024-11-02

**Soundness:** 2
**Presentation:** 3
**Contribution:** 2
**Rating:** 5
**Confidence:** 5

**Summary:**

This paper presents WinnowRAG, a training-free framework designed to address the challenges of utilizing a large volume of retrieved documents in RAG systems. WinnowRAG employs query-aware clustering and a multi-agent winnowing process to filter out noisy information in retrieved documents while retaining relevant content. Consequently, WinnowRAG improves the accuracy and relevance of generated responses without requiring model-specific fine-tuning. The experimental results highlight its potential as a robust method for integrating external knowledge into language models, offering insights for more reliable, contextual, and knowledge-intensive applications across various domains.

**Strengths:**

1. The paper features clear and effective figures, making the experimental results easy to interpret. Additionally, the method is precisely formulated, facilitating straightforward implementation.

2. The motivation is direct and broadly applicable to RAG frameworks. WinnowRAG combines query-aware clustering with multi-agent winnowing to effectively filter out noisy documents.

3. WinnowRAG is a practical, training-free solution for document filtering, enhancing the relevance of retrieved information.

**Weaknesses:**

1. Efficiency Concerns: WinnowRAG’s use of multiple LLM-based agents for winnowing introduces significant computational overhead compared to mainstream RAG baselines. The authors should discuss this issue in the experiments section.

2. Limited Novelty: The core methods of query-aware clustering and multi-agent winnowing resemble existing approaches [1,2,3,4], which limits the technical novelty of this work.

3. Alternative Methods: Many other approaches effectively manage large sets of retrieved documents, such as reranking [5] and truncation [6]. The paper would benefit from a discussion comparing these alternatives.

4. Additional Baselines: Using simpler filters to assess the utility of retrieved texts—leading to lower computational overhead and smaller models—is an important baseline that is missing from the discussion.

[1] Domain-aware Mashup Service Clustering Based on LDA Topic Model from Multiple Data Sources
[2] Improving Factuality and Reasoning in Language Models Through Multi-Agent Debate
[3] Examining Inter-Consistency of Large Language Models Collaboration
[4] MedAgents: Large Language Models as Collaborators for Zero-Shot Medical Reasoning
[5] Re2G: Retrieve, Rerank, Generate
[6] List-Aware Reranking-Truncation Joint Model for Search and Retrieval-Augmented Generation

**Questions:**

1. As shown in Figure 5, the number of clusters significantly impacts performance and is highly sensitive to the total number of documents. Why not consider other clustering methods, such as hierarchical clustering, which can dynamically adjust the number of clusters? Additionally, how is clustering performance evaluated before proceeding to the next step?

2. The merging step is not entirely clear. It is intended to filter documents, but what exactly do we achieve through this merging process? What is the relationship between two merged clusters? Including more examples in the paper would help clarify and demonstrate this strategy.

---

### Official Review · Reviewer_fHmx · 2024-11-04

**Soundness:** 3
**Presentation:** 3
**Contribution:** 2
**Rating:** 5
**Confidence:** 3

**Summary:**

This paper presents WinnowRAG, a retrieval-augmented generation framework designed to filter out lengthy and noisy documents while retaining valuable content. WinnowRAG begins by conducting query-aware clustering to organize the retrieved documents into clusters. Each cluster is then managed by an assigned LLM agent. The framework employs a critic agent to merge agents that provide similar answers into a super-agent using an ellipse merging strategy. Subsequently, several rounds of winnowing are conducted, wherein incorrect super-agents are merged with remaining agents using a hyperbola merging strategy. Finally, the critic agent concludes the winnowing and obtains the final answer. Extensive experiments across five datasets demonstrate the effectiveness of WinnowRAG.

**Strengths:**

1. The paper is well-organized and clearly written, making it easy to follow and understand.
2. The experiments are well-designed, effectively demonstrating the overall efficacy of the method and the effectiveness of its individual modules.

**Weaknesses:**

The primary concern is the efficiency of the method. The complex processing approach for retrieved documents, particularly the multi-agent winnowing with multiple iterations,  significantly increases inference costs and time, limiting the system's usability. There is a lack of experiments comparing and discussing the efficiency of the proposed WinnowRAG with existing methods.

**Questions:**

1. 2WikiMQA seems to be the dataset that would most benefit from an increased retrieval number, as its retrieval model has the lowest Recall@5 and Recall@20 in Table 1. However, as presented in Figure 4, increasing the number of retrieved documents did not yield the expected improvement for this dataset. Can you explain why this might be the case?
2. In the ablation study, merely removing the multi-round multi-agent winnowing module drastically reduced performance. As shown in Figure 3, "Ours w/o W" experiences a significant performance drop compared to "Ours," and its performance on the MHQA dataset appears inferior to Zero-shot Prompting.
3. There is an inconsistency between the naming of the MHQA dataset in the figures and the 2WikiMQA mentioned in the text.

---

### Note · Authors · 2024-12-16

I have read and agree with the venue's withdrawal policy on behalf of myself and my co-authors.